# Portraying entanglement between molecular qubits with four-dimensional inelastic neutron scattering

E. Garlatti[1], T. Guidi[2], S. Ansbro[3,4], P. Santini[1], G. Amoretti[1], J. Ollivier[3], H. Mutka[3], G. Timco[4], I.J. Vitorica-Yrezabal[4], G.F.S. Whitehead[4,†], R.E.P. Winpenny[4] & S. Carretta[1]

Entanglement is a crucial resource for quantum information processing and its detection and quantification is of paramount importance in many areas of current research. Weakly coupled molecular nanomagnets provide an ideal test bed for investigating entanglement between complex spin systems. However, entanglement in these systems has only been experimentally demonstrated rather indirectly by macroscopic techniques or by fitting trial model Hamiltonians to experimental data. Here we show that four-dimensional inelastic neutron scattering enables us to portray entanglement in weakly coupled molecular qubits and to quantify it. We exploit a prototype $(Cr_7Ni)_2$ supramolecular dimer as a benchmark to demonstrate the potential of this approach, which allows one to extract the concurrence in eigenstates of a dimer of molecular qubits without diagonalizing its full Hamiltonian.

[1] Dipartimento di Scienze Matematiche, Fisiche ed Informatiche, Università di Parma, I-43124 Parma, Italy. [2] ISIS facility, Rutherford Appleton Laboratory, OX11 0QX Didcot, UK. [3] Institut Laue-Langevin, 71 Avenue des Martyrs CS 20156, Grenoble Cedex 9 F-38042, France. [4] School of Chemistry and Photon Science Institute, The University of Manchester, M13 9PL Manchester, UK. † Present Address: Department of Chemistry, University of Liverpool, Liverpool L69 7ZD, UK. Correspondence and requests for materials should be addressed to S.C. (email: stefano.carretta@fis.unipr.it).

One of the most intriguing aspects of quantum mechanics are entangled states[1–3], which exhibit correlations that cannot be accounted for in classical physics. Entanglement occurs when a composite quantum object is described by a wave function that is not factorized into states of the object's components, making it impossible to describe a part of the system independently of the rest of it.

Recently, the concept of entanglement has been applied to many different areas of quantum many-body physics, including condensed matter[4,5], high-energy field theory[6] and quantum gravity[7]. In particular, the development of quantum information science[4,8] has largely increased the interest in the study of entanglement, since it represents an essential resource for quantum information processing applications. Thus, how to experimentally detect and quantify entanglement has become

a crucial step for the development of quantum information protocols. However, experimental measurements of entanglement in complex systems are very difficult, because quantum state tomography[9] requires demanding resources and a high degree of control[10,11]. Recently, many theoretical and experimental works have focused on the detection of entanglement[12–19].

Arrays of weakly coupled molecular nanomagnets represent an ideal playground for investigating quantum entanglement between complex spin systems[4,20]. In particular, supramolecular complexes containing linked antiferromagnetic rings have been demonstrated to be excellent candidates for implementing quantum-computation[21] and quantum-simulation[22] algorithms (see also Supplementary Note 1). A large number of supramolecular complexes of molecular nanomagnets are now being synthesized, but so far entanglement between molecular subunits has only been experimentally demonstrated by exploiting susceptibility as a witness in a $(Cr_7Ni)_2$ dimer or by fitting trial model Hamiltonians to electron paramagnetic resonance data[23]. The dimer (see Fig. 1a) consists of two $Cr_7Ni$ antiferromagnetic rings, and hence is a complex system made of 16 interacting spins, that is, one on each metal site. Strong exchange couplings between the eight spins within each ring lock these spins into a correlated $S = 1/2$ ground state[24], which is preserved in the presence of sizeable magnetic fields. Importantly, a weaker inter-ring interaction leads to entanglement between the two composite molecular $S = 1/2$ spins. The four-dimensional inelastic neutron scattering technique (4D-INS)[25] has the potential to provide a precise characterization of entanglement in such a system. Indeed, the crosssection directly reflects dynamical correlations between individual atomic spins in the molecule, and distinguishes between intra-ring correlations, associated with the composite nature of the molecular qubits, and inter-ring spin–spin correlations, which are a signature of entanglement between qubits.

Here we use this prototype $(Cr_7Ni)_2$ dimer as a benchmark system to demonstrate the capability of 4D-INS to investigate entanglement between molecular qubits. By assuming the rings maintain the $S = 1/2$ ground doublets determined from previous measurements on single-rings[26], the richness of the experimental information enables us to quantify inter-ring entanglement in the eigenstates through the determination of their concurrence. A magnetic field is used to place the system into a factorized ground state of the two rings and test the occurrence of entanglement in a comfortable parameter regime. Indeed, in this way we can univocally attribute the observed inter-ring correlations to the entanglement in the excited states. This is

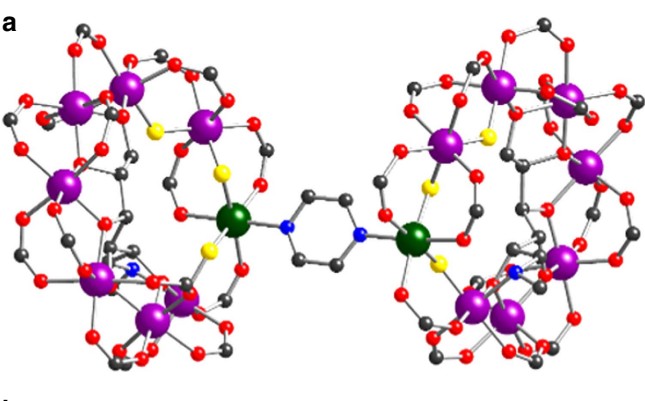

**a**

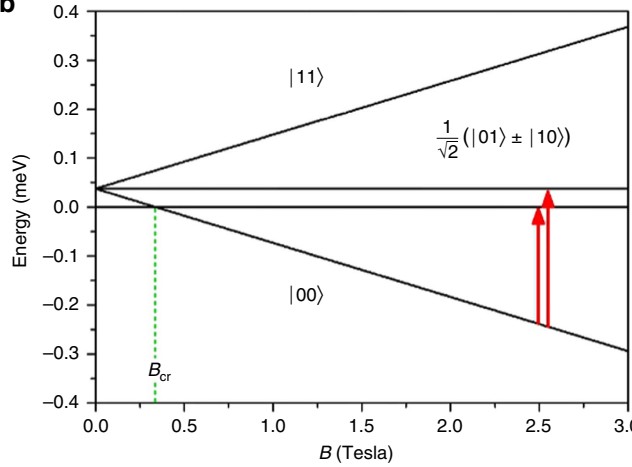

**b**

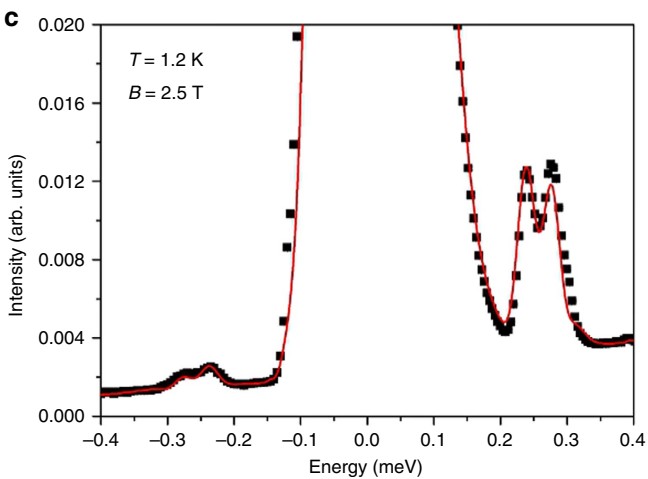

**c**

**Figure 1 | The $(Cr_7Ni)_2$ dimer. (a)** Molecular structure of the supramolecular dimer of formula $[Cr_7NiF_3(C_7H_{12}NO_5)(O_2CC(CH_3)_{15})]_2(N_2C_4H_4)$ (magenta, Cr; green, Ni; yellow, F; red, O; light blue, N; grey, C; $C(CH_3)_3$ groups and all H atoms omitted for clarity), consisting of two purple-$(Cr_7Ni)$ AF rings, forming an angle of about 30°. The coupling between the rings is mediated by pyrazine, which directly links the two Ni ions. **(b)** Lowest-energy levels of the molecular dimer as a function of the applied magnetic field B, calculated by diagonalizing the microscopic spin Hamiltonian, equations (1 and 2). The arrows indicate the two transitions detected by in-field INS. $|0\rangle$ and $|1\rangle$ qubit states are encoded in the ground $|S = 1/2, M = \pm 1/2\rangle$ doublet of each ring. Above $B_{cr} = 0.34\,T$ the dimer ground state is a factorized ferromagnetic state. In the first two excited states the total spins ($S = 1/2$) of the rings are entangled and correspond to two Bell states. **(c)** Measured INS spectrum (black points) and calculations (red line) for a collection of $(Cr_7Ni)_2$ crystals at $T = 1.2\,K$ and $B = 2.5\,T$ applied along the y-vertical axis, with an incident neutron wavelength $\lambda = 7.5\,\text{Å}$ and integrated over the full range of measured Q. Excitations in both energy-gain and energy-loss sides of the spectrum are clearly visible.

possible because of the strong intra-ring couplings which preserve the $S = 1/2$ ground states. This approach can be applied also to dimers of more complex molecular qubits (described as pseudo-spin 1/2) and it does not require any assumption on the form of the interaction between the qubits.

## Results

**The (Cr₇Ni)₂ supramolecular dimer.** The Cr₇Ni antiferromagnetic rings constitute the most studied family of molecular qubits[21,27–29]. The magnetic core is formed by seven Cr ions and one Ni ion arranged at the corners of an octagon. The dominant interaction is the nearest-neighbor antiferromagnetic exchange and leads to an isolated ground doublet behaving as a total spin $S = 1/2$, which can be used to encode a qubit. These molecules display coherence times sufficiently long for spin manipulations[29] and they have been proposed as prototype for implementing quantum gates[28,30] and quantum simulations[22].

Here we study the $[Cr_7NiF_3(C_7H_{12}NO_5)(O_2CC(CH_3)_{15})]_2$ $(N_2C_4H_4)$ supramolecular dimer, where $N_2C_4H_4 = N$-methyl-D-glucamine (Fig. 1a). The synthesis of (Cr₇Ni)₂ is described in the 'Methods' section. The Cr₇Ni subunits have been fully characterized by INS and electron paramagnetic resonance spectroscopies and by low-temperature specific heat and magnetometry measurements[26]. This characterization shows, unequivocally, that the ground state is a $S = 1/2$ doublet and that it is the only state of the ring populated at the temperatures used in the present work. Indeed, the first excited multiplet lies at about 18 K. The Hamiltonian describing each Cr₇Ni molecule is (assuming the Ni ion on site 8)

$$H = J \sum_{i=1}^{6} \mathbf{s}(i) \cdot \mathbf{s}(i+1) + J'[\mathbf{s}(1) \cdot \mathbf{s}(8) + \mathbf{s}(7) \cdot \mathbf{s}(8)]$$
$$+ \sum_{i=1}^{8} d_i s_z^2(i) - \mu_B \sum_{i=1}^{8} g_i \mathbf{B} \cdot \mathbf{s}(i), \tag{1}$$

where $\mathbf{s}(i)$ is the spin operator of the the $i$th ion ($s = 3/2$ for $Cr^{3+}$ and $s = 1$ for $Ni^{2+}$). The first two terms correspond to the dominant antiferromagnetic isotropic exchange interaction (with $J = 20$ K and $J' = 30$ K), while the third term describes the axial single-ion zero-field-splitting terms (with $d_{Cr} = -0.34$ K, $d_{Ni} = -7.3$ K and the $z$-axis perpendicular to the ring). The last term represents the Zeeman coupling with an external field $B$ (with $g_{Cr} = 1.98$ and $g_{Ni} = 2.1$). The two Cr₇Ni rings are linked through a pyrazine unit, which provides two N-donor atoms binding to Ni centres in different rings. This leads to a weak exchange coupling between the Ni ions:

$$H_{int} = j \, \mathbf{s}(Ni_A) \cdot \mathbf{s}(Ni_B), \tag{2}$$

where $A$ and $B$ label the two rings. We have checked that the effects of the anisotropic zero-field-splitting terms on the results presented in this work are very small (see Supplementary Fig. 1), hence hereafter we neglect the third term in equation (1).

Figure 1b reports the calculated magnetic field dependence of the lowest-energy levels of (Cr₇Ni)₂. In the presence of an antiferromagnetic ring–ring coupling ($j > 0$), the supramolecular system is characterized by an entangled singlet ground state in zero applied field and by an excited triplet. Within this subspace each ring behaves as a total spin $S = 1/2$, because the strong intra-ring exchange interactions rigidly lock together the individual spins within each ring. This condition is preserved also in sizeable fields, which can then be used to induce (for $B > B_{cr}$) a factorized ferromagnetic ground state for the dimer. In this regime, by focusing on a specific low-temperature transition, we can selectively investigate the entanglement between the rings in the corresponding excited state. Indeed, the origin of the observed

inter-ring correlations can be univocally attributed to the excited state involved in the transition (see below) because the ground state is factorized. Moreover, the application of a magnetic field significantly improves the experimental working conditions for detecting inter-ring interactions. In fact, it is easier to resolve a very small splitting $\delta$ between two peaks in the centre of the energy-transfer range (as it occurs in a sizeable field), than to observe a single-peak centred at $\delta$ (zero-field case), because it would be partially covered by the elastic signal.

**Portraying entanglement with 4D INS.** The 4D-INS technique[25] has the potential to yield a deep insight into the entanglement between molecular qubits, by measuring the 4D scattering function $S(\mathbf{Q}, \omega)$ in large portions of the reciprocal $\mathbf{Q}$ space and in the relevant energy-transfer $\hbar\omega$ range. Indeed, the dependencies of the transition intensities on $\mathbf{Q}$ yield direct information on the dynamical spin–spin correlation functions[25]. The zero-temperature spin scattering function is ref. 31

$$S(\mathbf{Q}, \omega) \propto \sum_{\alpha,\beta=x,y,z} \left( \delta_{\alpha,\beta} - \frac{Q_\alpha Q_\beta}{Q^2} \right) \sum_{p,d,d'} F_d(Q) F_{d'}(Q)$$
$$\exp(i\mathbf{Q} \cdot \mathbf{R}_{dd'}) \langle 0|s_\alpha(d)|p\rangle \langle p|s_\beta(d')|0\rangle \delta(\omega - E_p/\hbar), \tag{3}$$

where $F_d(Q)$ is the magnetic form factor for the $d$th ion, $|0\rangle$ and $|p\rangle$ are ground and excited eigenstates, respectively, $E_p$ are eigenenergies and $\mathbf{R}_{dd'}$ are the relative positions of the $N$ magnetic ions within the supramolecular dimer. The products of spin matrix elements are the Fourier coefficients of $T = 0$ dynamical correlation functions

$$\langle s_\alpha(d,t)s_\beta(d',0)\rangle = \sum_p \langle 0|s_\alpha(d)|p\rangle \langle p|s_\beta(d')|0\rangle e^{-iE_p t/\hbar}. \tag{4}$$

If $|0\rangle$ is a factorized reference state, and $|p\rangle$ is an excited state where the two rings are entangled, the dynamical correlations of equation (4) are non-zero also for pairs of spins where $d$ is in one ring and $d'$ in the other one. Conversely, these inter-ring correlations would be zero if the states of the two monomers were factorized also in $|p\rangle$, because the corresponding products of spin matrix elements would vanish. The spatial structure of these large-distance correlations produces short-$Q$ modulations in $S(\mathbf{Q}, E_p/\hbar)$ through the $\exp(i\mathbf{Q} \cdot R_{dd'})$ factors in equation (3), see Fig. 2. Hence, the corresponding pattern of maxima and minima in the measured $S(\mathbf{Q}, E_p/\hbar)$ is a portrayal of the entanglement between molecular qubits in state $|p\rangle$.

We have measured by 4D-INS a collection of (Cr₇Ni)₂ single crystals, exploiting the position-sensitive-detectors of the cold-neutron time-of-flight spectrometer IN5 at the Institute ILL in Grenoble[32]. This kind of experiment is very challenging because of two conflicting requirements: on the one hand, a detailed $\mathbf{Q}$-dependence of the scattering function over a large range of $\mathbf{Q}$ is needed. On the other hand, very high resolution is needed to resolve INS transitions within the dimer's lowest-energy manifold, whose splittings are due to the small inter-ring coupling. In addition, the large single crystals of molecular nanomagnets required for INS are very difficult to grow. To guarantee the occurrence of a factorized reference (ground) state we have applied a magnetic field $B = 2.5$ T, much stronger than any ring–ring interaction. In addition, measurements have been performed at $T = 1.2$ K to make the populations of excited states negligible. From the model of equations (1) and (2) two peaks are expected, corresponding to excitations from the factorized ground state towards two excited Bell states (marked by red arrows in Fig. 1b). Figure 1c shows that two inelastic peaks are actually observed at 0.24 and 0.28 meV, both in the energy gain and energy-loss sides (Fig. 1c). These findings are

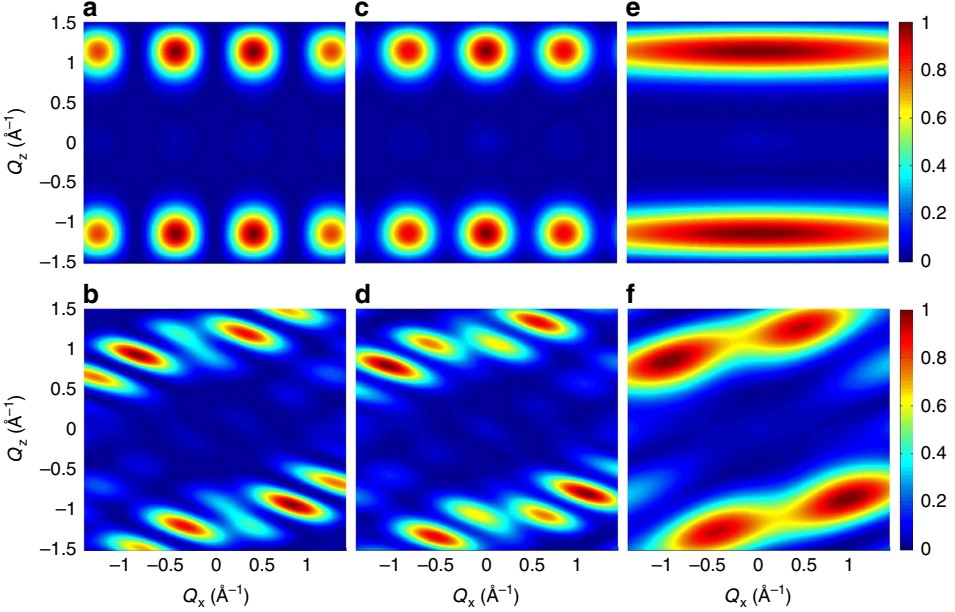

**Figure 2 | Effects of inter-ring correlations on the intensity maps.** Calculated neutron scattering intensity for the two in-field low-temperature inelastic transitions of the $(Cr_7Ni)_2$ dimer in the $Q_x$-$Q_z$ plane. The cross section has been integrated over a symmetric $Q_y$ range from $-0.2$ to $0.2\,\text{Å}^{-1}$ and over energy ranges centred around the observed transition energies: $0.2\,\text{meV} < E < 0.25\,\text{meV}$ for **a,b** and $0.25\,\text{meV} < E < 0.33\,\text{meV}$ for **c,d**. The colour bar reports the transition intensity normalized for the maximum in each panel. The magnetic field $B = 2.5\,\text{T}$ is applied along the y-vertical axis. Calculations are shown in **a** and **c** for a single ideal dimer and in **b,d** for the real one with non-parallel rings. Short-$Q$ modulations disappear when inter-ring correlations are forced to zero in equation (3) (**e** for an ideal dimer and **f** for the real system).

reproduced by equations (1) and (2) with $j = 1.1\,\text{K}$. Preliminary measurements were also performed on the high-resolution spectrometer LET at ISIS to estimate the ring–ring coupling.

To illustrate the features in $S(\mathbf{Q}, \omega)$ reflecting inter-ring correlations and entanglement, consider the ideal case of a dimer composed by two perfectly regular and parallel rings, lying in planes parallel to the y–z plane. In this case all intra-ring distance vectors $\mathbf{R}_{dd'}$ are in the y–z plane, whereas inter-ring vectors have a large component along x (axis perpendicular to the rings). Thus, modulations of $S(\mathbf{Q}, \omega)$ as a function of $Q_x$ directly reflect inter-ring dynamical correlations and entanglement through the term $\exp(i\mathbf{Q} \cdot \mathbf{R}_{dd'})$ in equation (3). Conversely, intra-ring correlations lead to modulations of $S(\mathbf{Q}, \omega)$ as a function of $Q_y$ and $Q_z$. This is illustrated in Fig. 2a,c, that report the calculated INS intensity in the $Q_x$–$Q_z$ plane for the two transitions in Fig. 1c, for an ideal $(Cr_7Ni)_2$ dimer described by equations (1) and (2). $S(\mathbf{Q}, \omega)$ is characterized by several maxima and minima whose pattern depends on the specific transition and reflects the structure of the involved wavefunctions. In particular, these maps clearly display $Q_x$-dependent modulations, due to the presence of entanglement in the excited $|p\rangle$ states of the dimer. On the contrary, if correlations between the two rings are forced to zero in equation (3), $Q_x$-dependent modulations disappear (Fig. 2e). The residual smooth $Q_x$-dependency of the intensity is merely due to single-ion form factors. A similar behaviour is observed if an Ising inter-ring interaction is considered, as it leads to factorized states (see Supplementary Note 3 and Supplementary Fig. 2). Figure 2b,d,f report the same calculations for a real $(Cr_7Ni)_2$ dimer, in which the planes of the rings are not parallel (they form an angle of about $30°$) and their normals do not coincide with any axis of the laboratory reference frame (see 'Methods' section). Short-$Q$ modulations of the intensity are still clearly visible for the two transitions to the entangled $|p\rangle$ states of the dimer (Fig. 2b,d) and are absent in Fig. 2f. Longer-period modulations are due to intra-ring correlations in the two non-parallel rings.

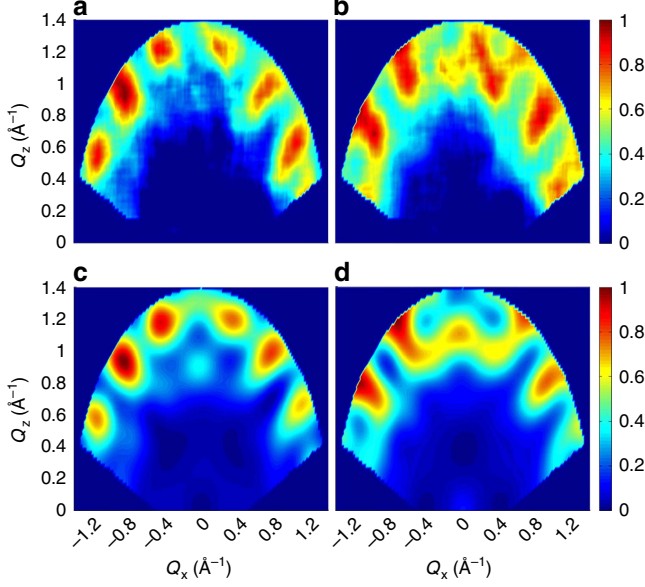

**Figure 3 | Constant-energy plots of the neutron scattering intensity.** **a,b** show the observed dependency of the neutron scattering intensity of the two inelastic excitations on the two horizontal wavevector components $Q_x$-$Q_z$, integrated over the full experimental $Q_y$ data range measured on IN5 of $-0.2$ to $0.2\,\text{Å}^{-1}$. The measured cross section has been integrated over energy ranges centred around the observed transition energies: $0.2\,\text{meV} < E < 0.25\,\text{meV}$ for **a** and **c** and $0.25\,\text{meV} < E < 0.33\,\text{meV}$ for **b,d**. The colour bar reports the transition intensity normalized for the maximum in each panel. The measurement was carried out with a $7.5\,\text{Å}$ incident neutron wavelength, a sample temperature of $1.2\,\text{K}$ and a magnetic field $B = 2.5\,\text{T}$ applied along the y-vertical axis. **c,d** report the corresponding calculations accounting for the presence of differently oriented dimers in the crystals.

Figure 3a,b report the measured **Q**-dependence of the intensity of the two observed transitions. Short-$Q$ modulations of the intensity are evident for both transitions, with the maxima in the low-energy excitation corresponding to minima in the high-energy one and vice versa. The observed pattern is more complex than in Fig. 2 because dimers with different orientations are present in the single crystals (see 'Methods' section). Nevertheless, these results clearly demonstrate the occurrence of entanglement in the excited states of the supramolecular dimer. Indeed, Fig. 3c,d show that the predicted short-$Q$ modulations coincide with the experimental findings. It is worth stressing that magnetic anisotropy plays a negligible role in these results (see Supplementary Note 2), hence the different orientations of the dimers in the crystals do not affect the possibility of demonstrating the occurrence of entanglement.

**Quantification of the entanglement between molecular qubits.** Having experimentally shown the occurrence of entanglement, the next important question is whether is it possible to quantify it. In the following we show that it is possible to extract the concurrence $C$ from the portrait reported in Fig. 3. $C$ is one of the most used measures of the entanglement between a pair of qubits[33], and its value ranges from 0 for factorized states to 1 for maximally entangled ones. For pure two-qubit states $|p\rangle = a|00\rangle + b|10\rangle + c|01\rangle + d|11\rangle$, the concurrence is

$$C = 2|ad - bc|. \tag{5}$$

It is evident that $C = 0$ for a factorized state like $|01\rangle$ or $|10\rangle$ and $C = 1$ for the Bell states $(|10\rangle \pm |01\rangle)/\sqrt{2}$.

In the case of a molecular qubit, the $|0\rangle$ and $|1\rangle$ states are typically encoded in the two lowest-energy eigenstates (for example, the ground total spin $S = 1/2$ doublet in $Cr_7Ni$). By considering two molecular qubits ($A$, $B$) and restricting to the computational basis (that is, the subspace in which both molecular qubits are in the lowest doublet), the scattering function $S(\mathbf{Q}, E_p/\hbar)$ for the transitions from the factorized $|00\rangle$ ground state to excited $|p\rangle$ states contains two single-qubit contributions and an interference term:

$$S(\mathbf{Q}, E_p/\hbar) = |b|^2 I_{AA}(\mathbf{Q}) + |c|^2 I_{BB}(\mathbf{Q}) + 2Re[b\,c^* I_{AB}(\mathbf{Q})]. \tag{6}$$

By considering molecular qubits characterized by a well isolated $S = 1/2$ doublet, the explicit expression of the three contributions is:

$$I_{AA} = \sum_{\alpha,\beta=x,y,z} \left( \delta_{\alpha,\beta} - \frac{Q_\alpha Q_\beta}{Q^2} \right) \sum_{d,d'\in A} F_d(Q) \tag{7}$$

$$F_{d'}(Q)\exp(i\mathbf{Q}\cdot\mathbf{R}_{dd'})c_d c_{d'}\langle 00|S_\alpha^A|10\rangle\langle 10|S_\beta^A|00\rangle$$

$$I_{BB} = \sum_{\alpha,\beta=x,y,z} \left( \delta_{\alpha,\beta} - \frac{Q_\alpha Q_\beta}{Q^2} \right) \sum_{d,d'\in B} F_d(Q) \tag{8}$$

$$F_{d'}(Q)\exp(i\mathbf{Q}\cdot\mathbf{R}_{dd'})c_d c_{d'}\langle 00|S_\alpha^B|01\rangle\langle 01|S_\beta^B|00\rangle$$

$$I_{AB} = \sum_{\alpha,\beta=x,y,z} \left( \delta_{\alpha,\beta} - \frac{Q_\alpha Q_\beta}{Q^2} \right) \left[ \sum_{d\in A,d'\in B} F_d(Q) \tag{9} \right.$$

$$\left. F_{d'}(Q)\exp(i\mathbf{Q}\cdot\mathbf{R}_{dd'})c_d c_{d'}\langle 00|S_\alpha^A|10\rangle\langle 01|S_\beta^B|00\rangle \right],$$

where the second summation is over the sites $d$ and $d'$ of ring $A$ or $B$, $S_\alpha^{A,B}$ are effective spin-1/2 operators acting in the ground doublet of each molecular qubit and $c_d$ are the corresponding projection coefficients[34]. This formula can be generalized to other types of molecular doublets. The factorized $|00\rangle$ ground state (obtained by applying a sizeable magnetic field) implies that $a = 0$ because $\langle 00|p\rangle = 0$.

All the quantities in $I_{AA}$, $I_{BB}$ and $I_{AB}$ are here calculated by assuming the $S = 1/2$ doublet deduced from the model interpreting measurements on the single-ring compound[26]. It is important to note that these quantities can be also determined from measurements on single-qubit compounds (for instance, in $Cr_7Ni$ the $c_d$ coefficients can be extracted by measuring local magnetic moments with nuclear magnetic resonance[35] or polarized neutron diffraction[36]). Hence, the concurrence $C = 2|bc|$ of the two excited states can be extracted from the data by fitting the observed **Q**-dependence (Fig. 3) with equation (6). It is worth noting that the inter-qubit interference contribution to $S(\mathbf{Q}, E_p/\hbar)$ (third term in equation (6)) enables one to discriminate between $|p\rangle$ states with real and complex coefficients. For the present experimental configuration, the observed short-$Q$ modulations in $S(\mathbf{Q}, E_p/\hbar)$, integrated over an asymmetric range of $Q_y$, point to real wavefunctions (see Supplementary Note 4 and Supplementary Fig. 3), as expected for dominant Heisenberg interactions. Figure 4 shows the calculated **Q**-dependence of the transitions assuming different real values of $b$ and $c$, corresponding to different values of $C$. Calculations for

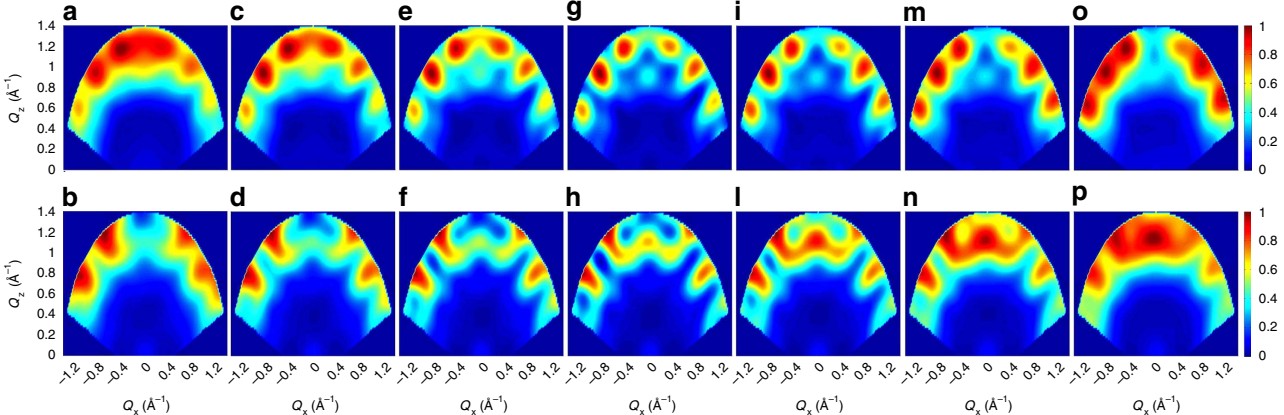

**Figure 4 | Intensity plots for different values of the concurrence C.** Calculations of the dependency of the INS intensity on the wavevector components $Q_x$–$Q_z$ (integrated over the measured $Q_y$ range) for different compositions of the excited states $|p\rangle$ (assuming $d = 0$ because of the sizeable applied field). The colour bar reports the transition intensity normalized for the maximum in each panel. Figures in the first (second) row correspond to the lower-energy (higher-energy) excitation. Real wavefunctions with positive values of $c$ increasing (decreasing) from left to right are shown in the first (second) row and lead to different values of the concurrence: $C = 0.25$ for **a,b,o,p**, $C = 0.5$ for **c,d,m,n**, $C = 0.75$ for **e,f,i,l** and $C = 1$ for **g,h**.

positive values of $c$ increasing (decreasing) from left to right are reported in the first (second) row, while examples with complex wavefunctions are reported in the Supplementary Note 4. The comparison between Figs 3a,b and 4 clearly shows that $C \simeq 1$ for both excited states. It is worth noting that this kind of information cannot be extracted by measuring the energy spectrum as in Fig. 1c, because the mere observation of two peaks is compatible with many possible models with $C < 1$. For instance, an Ising dimer model with two slightly different $g$ values would yield a similar INS energy spectrum, but the two excited states would be factorized and $C = 0$ (see Supplementary Fig. 2). Furthermore, by determining the concurrence of the eigenstates by this approach, it is also possible to extract the concurrence in the thermodynamic equilibrium state as a byproduct[37].

## Discussion

To summarize, by using the $(Cr_7Ni)_2$ supramolecular dimer as a benchmark, we have shown that the richness of 4D-INS enables us to portray entanglement in weakly coupled molecular qubits and to quantify it. This possibility opens remarkable perspectives in understanding of entanglement in complex spin systems. Molecular nanomagnets are among the best examples of real spin systems with a finite size, and are an ideal testbed to address this issue: they are also currently attracting increased attention for quantum information processing[38–40]. Tailored finite spin systems can also be assembled on a surface by using a scanning tunnelling microscope[41,42], but molecular nanomagnets have the advantage that a macroscopic number of identical and independent units can be gathered in the form of high-quality crystals. These enable elusive properties like entanglement to be explored by bulk techniques as 4D INS. It is important to underline that the present method works independently of the specific form of the inter-qubit interaction. Indeed, neither the demonstration of the entanglement through the observed short-$Q$ modulations of the intensity, nor its quantification using equation (6), exploit equation (2). Hence, entanglement can be investigated even between molecular qubits where a sound determination of the full spin Hamiltonian is not possible, as might be the case for molecules containing $4f$ or $5f$ magnetic ions.

The use of larger single crystals and the consequent increase in the quality of the data, together with additional measurements at higher energies, will even allow the determination of two-spin dynamical correlation functions involving individual spins belonging to different molecular qubits. For instance, this would permit the direct determination of the quantum Fisher information[13], a witness for genuinely multipartite entanglement and the detection of entanglement between subsystems of the complex spin cluster[15]. Finally, the improvement in the flexibility and in the flux of the new generation of spectrometers, like those under development at the new European Spallation Source (https://europeanspallationsource.se), will further expand the possibilities of this kind of technique.

## Methods

**Synthesis and crystal structure.** Unless stated otherwise, all reagents and solvents were purchased from Sigma-Aldrich and used without further purification. Analytical data were obtained by the Microanalysis laboratory at the University of Manchester. Carbon, hydrogen and nitrogen analysis (CHN) by Flash 2000 elemental analyser. Metals analysis by Thermo iCap 6300 Inductively coupled plasma optical emission spectroscopy.

Compound $[Cr_7NiF_3(Meglu)(O_2C^tBu)_{15} \cdot Et_2O]$ (1) was obtained by a similar method reported in ref. 43 for $Cr_7NiF_3(Etglu)(O_2C^tBu)_{15} \cdot Et_2O]$ in $\sim 30\%$ yield by using $N$-methyl-$D$-glucamine ($H_5$Meglu $= C_7H_{12}NO_5H_5$) instead of $N$-ethyl-$D$-glucamine ($H_5$Etglu $= C_8H_{14}NO_5H_5$).

Synthesis of $[Cr_7NiF_3(Meglu)(O_2C^tBu)_{15}]_2(C_4H_4N_2)$ (where $C_4H_4N_2 =$ pyrazine): pyrazine (0.1 g, 1.25 mmol) was added to a warm ($\sim 30\,^\circ C$) solution of 1 (6.0 g, 2.65 mmol) in dichloromethane anhydrous (130 ml) and the solution was stirred for 5 min, and then allowed to stand at room temperature for 7 days, during which

time dark purple crystals of $[Cr_7NiF_3(Meglu)(O_2C^tBu)_{15}]_2$ $(C_4H_4N_2)$ grew. The crystals were collected by filtration, washed with a small amount of dichloromethane and dried in a flow of nitrogen. Yield: 4.3 g (77%, based on pyrazine); elemental analysis calculated (%) for $(C_{168}H_{298}Cr_{14}F_6N_4 Ni_2O_{70})$: Cr 16.35, Ni 2.64, C 45.31, H 6.75, N 1.26; found: Cr 15.89, Ni 2.53, C 45.81, H 6.77, N 1.07.

Single-large crystals preparation of $(Cr_7Ni)_2$ dimer: a powder of $[Cr_7NiF_3(Meglu)(O_2C^tBu)_{15}]_2(C_4H_4N_2)$ (3.0 g) was dissolved in refluxing dichloromethane anhydrous (100 ml) in an Erlenmeyer 500-ml flask, while stirring for 10 min and then anhydrous anisole ($C_7H_8O$) (25 ml) added and the solution was filtered. The filtrated was left undisturbed at ambient temperature under nitrogen for 2 weeks, during which time large well-shaped needles crystals alongside with small crystals including good X-ray quality crystals grew. The crystals were identified by X-ray crystallography as $[Cr_7NiF_3(Meglu)(O_2C^tBu)_{15}]_2$ $(C_4H_4N_2) \cdot (CH_2Cl_2) \cdot 4(C_7H_8O)$ in short $(Cr_7Ni)_2$ dimer, and maintained in contact with the mother liquor to prevent degradation of the crystal quality. Elemental analysis for a powder sample obtained from several large $(Cr_7Ni)_2$ dimer crystals dried *en vacuo*; elemental analysis calculated (%) for $(C_{168}H_{298}Cr_{14}F_6N_4Ni_2O_{70})$: Cr 16.35, Ni 2.64, C 45.31, H 6.75, N 1.26; found: Cr 16.18, Ni 2.58, C 45.24, H 6.73, N 1.14.

X-ray data for compound $[Cr_7NiF_3(C_7H_{12}NO_5)(O_2CC(CH_3)_3)_{15}]_2]_2(N_2C_4H_4)$ were collected at a temperature of 150 K using a using Mo–K radiation on an Agilent Supernova, equipped with an Oxford Cryosystems Cobra nitrogen flow gas system. Data were measured and processed using CrysAlisPro suite of programs. More details on the crystal structure determination, crystal data and refinement parameters are reported in Supplementary Note 5 and Supplementary Table 1.

Supplementary Data 1 contains the supplementary crystallographic data for this paper.

**Neutron scattering experiments.** INS experiments were performed on the IN5 time-of-flight inelastic neutron spectrometer[32] at the high-flux reactor of the Institute Laue-Langevin. The IN5 instrument has a 30 $m^2$ position sensitive detector divided in $10^5$ pixels, covering 147° of azimuthal angle and $\pm 20°$ out-of-plane. Six $(Cr_7Ni)_2$ single crystals were aligned on a copper sample holder with the [010] direction vertical. The crystals were placed with the flat faces perpendicular to the [110] or [1–10] directions lying on the sample holder, giving two set of crystal orientations. The dimensions of the crystals ranged from a minimum of $5 \times 3 \times 2$ mm to a maximum of $8 \times 4 \times 2$ mm. Measurements were taken by rotating the crystals (in steps of 1°) about the vertical axis, labelled $y$ in the laboratory frame.

An incident neutron wavelength of 7.5 Å was selected to probe the excitations centred at 0.24 and 0.28 meV, with an energy resolution (full-width at half-maximum) of 22 $\mu eV$ at the elastic line. The magnetic field was applied along the $y$-vertical axis.

Preliminary measurements were also performed on the high-resolution LET[44] spectrometer at ISIS neutron spallation source using an incident energy of 2.5 meV (40 $\mu eV$ energy resolution), at a temperature of 1.8 K and a field of 5 T.

**Data analysis and simulations.** Measurements for different rotation angles were combined using the HORACE analysis suite[45]. Corrections for attenuation as a function of rotation and scattering angle based on the slab-like geometry of the sample holder were performed using the formula given in ref. 46. To isolate magnetic signals from the background and to separate in energy the two transitions, the data reported in Fig. 3a,b have been obtained by integrating over the full $Q_y$ range and by fitting, for each set $(Q_x, Q_z)$, the resulting energy dependence with two gaussians plus a background contribution. The four supra-molecules in the unit cell were included in the calculation of the scattering functions in equations (3) and (6). The two sets of crystal orientations were also taken into account in our simulations, to obtain the intensity maps in Figs 3c,d and 4.

**Data availability.** Supplementary Data 1 has been deposited at the Cambridge Crystallographic Data Centre (deposition number: CCDC 1480868) can be obtained free of charge via www.ccdc.cam.ac.uk/conts/retrieving.html (or from the Cambridge Crystallographic Data Centre, 12 Union Road, Cambridge CB21EZ, UK; fax: ( + 44)1223-336-033; or deposit@ccdc.cam.ac.uk).

Raw data from the INS experiment were generated at the Insitut Laue-Langevin large-scale facility and are available at the ILL Data Portal with the identifier http://doi.ill.fr/10.5291/ILL-DATA.4-04-457. Derived data supporting the findings of this study are available from the corresponding author upon reasonable request.

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

## Acknowledgements

Very useful discussions with M.L. Baker and F. Troiani are gratefully acknowledged. E.G., P.S., G.A. and S.C. acknowledge financial support from the FIRB Project No. RBFR12RPD1 of the Italian Ministry of Education and Research. G.T., I.J.V.-Y., G.F.S.W. and R.E.P.W. thank the EPSRC(UK) for support, including funding for an X-ray diffractometer (grant number EP/K039547/1) and for the CDT NowNANO, which supported G.F.S.W. We acknowledge the Institute Laue-Langevin for funding the PhD of S.A. and for instrument time. We thank the ILL technical staff, in particular S. Turc and J. Halbwachs for technical assistance during the experiment. We thank John Crawford and Michael Hellier from the ISIS technical support for designing and machining the sample holders for the INS experiments.

## Author contributions

E.G., T.G., S.A., P.S., J.O., H.M. and S.C. performed the experiment on crystals synthesized by G.T. after discussion with R.E.P.W. Data treatment was made by E.G., T.G., S.A., J.O., H.M., S.C., and data simulations and fits were performed by E.G., P.S., G.A. and S.C. The structure of the compound has been determined by T.G., I.J.V.-Y. and G.F.S.W. E.G., T.G., P.S., G.A. and S.C. developed the idea to use 4D INS measurements to portray and quantify entanglement in dimers of molecular nanomagnets. E.G., P.S., G.A. and S.C. also did theoretical calculations. E.G. and S.C. wrote the manuscript with inputs from all coauthors.

## Additional information

**Competing financial interests:** The authors declare no competing financial interests.

**Publisher's note**: 

