## [Peer Review File · Nature Communications]

Reviewers' comments:

Reviewer #1 (Remarks to the Author):

Since the pioneer work of Wiesniak, Vedral and Brukner (ref. 14) on the quantification of entanglement in solid state physics by means of spin susceptibility and specific heat experiments, quite a number of compounds displaying antiferromagnetic $s=1/2$ dimer physics have been characterised. The ground state of these dimers is a non-magnetic spin singlet and, in the absence of anisotropy, a degenerate magnetic triplet arises as the excited state. The dimer systems is indeed a textbook example of entanglement where the proper states are the maximally entangled Bell states. The degeneracy of this triplet can be raised by the application of a magnetic field. Beyond a critical field, B_{cr} , the ground state changes and it can adopt different states, ranging from a Bose-Einstein condensation to a modulated magnetic structure. Importantly, in most of these compounds decoherence mechanisms act at the approach of B_{cr} and entanglement disappears beyond B_{cr} . The study of entanglement in AF spin $1/2$ dimer compounds at sufficiently low temperatures and magnetic fields -below B_{cr} - is a rather settled issue.

The manuscript by Garlatti et al is an interesting paper that reports on the quantification of entanglement by means of inelastic neutron scattering in supramolecular spin rings. I have to admit that my first reading of the manuscript left me puzzled and prompted me with very strange questions

- * why do the authors want to verify the occurrence of entanglement in weakly coupled antiferromagnetic rings forming a dimer by means of inelastic neutron scattering (INS) if entanglement has been already well characterised in this compound and published in ref. 23 ?
- * Why do the authors try to verify that in a regime of magnetic fields (2.5T) and temperature (1.2K) where concurrence should be zero, as it was stated in ref. 23 ?
- * And if so what have they measured in their INS ?

It could not be possible that the authors have made such mistakes. By this introduction I mean that the message conveyed in this manuscript is not clear to this reviewer, that the reader needs to go over the previous literature of supramolecular compounds in order to grasp the main ideas.

It was only after reading it a few times that I realised the utmost goal of this work where the key concept appears in the first lines of page 3 (and in ref. 23) and then I found it very interesting and certainly worth publishing in Nature Communications. The underline idea is that entanglement in these compounds is protected from decoherence by the rather strong magnetic exchange interactions that gives rise to a molecular $S=1/2$ in each ring. The effective exchange in the ring is 20-30K. This is to be compared with AF ring-ring exchange, 0.16K, that leads to a transition at round 50 mK and B_{cr} of 60mT (see fig. 4 in ref. 23). Therefore one can use a magnetic field to place the system into a factorised 00 ground state and test the occurrence of entanglement in a more comfortable parameter space of (T,B) . If this is the argument, please state it clearly !!! If not...

Building upon a previous work (ref. 24) authors have carried out INS experiments on a time-of-flight (TOF) instrument, where every detector records not only a Q-position but also the neutron energy by measuring the time a neutron takes to travel a given distance (that means neutron velocity and hence energy). By collecting this information in some hundreds of detectors, and at different single crystal orientations, one can produce a 4D maps of the excitations. And from that map and the use of a suitable software one can extract appropriate cuts of the excitations, as it is shown in the constant energy cuts in Fig. 3.

The next step is to perform simulations of the intensity contours from the scattering cross sections, including interference terms and concurrence coefficients, from the factorised state to the 01 and 10 states. The results of the simulation compares well with the experimental results. In particular Figs. 3 and 4 highlight the different intensity profiles of the $(01 + 10)$ and the $(01 - 10)$ states.

In conclusion this manuscript is a very interesting piece of work that certainly deserves to be published in Nature Communications albeit with substantial refurbishment and improved writing. I strongly advise the authors to

- Make use of the Supplement Information
- State clearly why this work is important and hence preventing the reader to go back to original publications to figure it out. This is the main caveat of this manuscript.
- In page 5 it has been made reference to the Ising model. Authors, could you please include such calculations in the Supplementary Information in order to strengthen their case ?

J.E. Lorenzo

Reviewer #2 (Remarks to the Author):

Authors have used data from neutron scattering measurements to detect the entanglement of spins located on different molecular spin rings that form a dimer. Novelty of this experiment comes from the clear identification of the features in the distribution of scattered neutrons that originate from the response of spins located on different rings. This analysis supports the claim that the distant spins are indeed entangled, without the need to perform a full state tomography which would be a forbiddingly complex task.

As the authors explained in the introduction, the entanglement is both crucial resource for implementation of quantum information processing and an important quantity that characterizes dynamics of interacting quantum systems. Dimers of Cr₇Ni rings used in this measurements have an important property that their spins show structure both at the level of isolated rings that leads to $S=1/2$ states and at the level of coupled frozen spin- $1/2$ states on the two rings. It is the entanglement between the rings that is important for the scaled-up application of the molecular spin rings.

The results presented in the manuscript give strong evidence that the spins on two monomers are indeed entangled. The technique used to support this claim is an extension of the analysis of inelastic neutron scattering off the compound similar to the monomers used in the current manuscript, where it was used to deduce the spin dynamics. As opposed to previous work, the new structure is clearly separated into total spin- $1/2$ monomers and the interaction induced correlation between these collective spins. The analysis is rather complete, the explanation of the methods is clear and simple, and the evidence for the basic claim is convincing.

What I find a bit less clear is the discussion of the results suggesting that the presented results are a direct measurement of the entanglement between the molecular rings which is independent of the details of the models used to describe them. After careful reading the true extent of the claim becomes apparent, namely that it is only the entanglement between individual spins on the monomer rings that is being directly measured. Entanglement between monomers is proven only after invoking a model. Therefore the authors should discuss the following remarks and restate their claims to make this point clearer:

1) The claim at the very end of introduction that the concurrence of the state of two spin is directly quantified is dependent on the validity of the models used in the analysis. Even though the used spin models are supported by many measurements and fit the data well it is not possible to claim that the measurement of the concurrence is direct. Similar issue appears in the analysis of Eq. 6, and the term $I_{\{AB\}}$ in it. Though the model with high concurrence shows the best fit to the observed data, the interpretation still depends on the facts that the monomer rings have total spin- $1/2$ states both in 0 and in p state. Therefore the language of the claim should explicitly state

this.

2) In the scattering intensity plots, the structure that is claimed to appear due to entanglement of monomers is in fact due to the correlation of spins at relatively large distances. Therefore it can only be claimed directly that the oscillatory structure is due to entanglement between elementary spins, on Cr and Ni atoms, that lie on different monomers. It is not clear that the entanglement is between the frozen spins $1/2$ and it is not clear that the concurrence of the state can be found until the assumption of total spin $1/2$ on each of the rings is invoked. Therefore the claim of high concurrence is dependent on the model, and it should be stated so.

There are more minor issues that should be resolved. The plots in Fig. 2 seem to have wrong labels. The plots do not correspond to the figure caption and the description in the main text. There is a mention of the states with artificially "turned off" correlations. The procedure for turning off the correlations should be explained explicitly.

Since the topic of the manuscript is rather interesting, and the results are important, well explained and well supported, I would recommend the manuscript for publication after these remarks are addressed.

Reviewer #1:

We thank Reviewer for his comments, which have drawn our attention to some weaknesses in the presentation of our results and helped us to improve the paper. Please find below a point-by-point response to each comment. A complete list of all the manuscript revisions is reported at the end of the document (after the response to Reviewer 2).

a) "It was only after reading it a few times that I realised the utmost goal of this work where the key concept appears in the first lines of page 3 (and in ref. 23) and then I found it very interesting and certainly worth publishing in Nature Communications. The underline idea is that entanglement in these compounds is protected from decoherence by the rather strong magnetic exchange interactions that gives rise to a molecular $S=1/2$ in each ring. The effective exchange in the ring is 20-30K. This is to be compared with AF ring-ring exchange, 0.16K, that leads to a transition at round 50 mK and B_{cr} of 60mT (see fig. 4 in ref. 23)."

The Referee is completely right: a central point of our work is that we are investigating entanglement in a complex system made of sixteen interacting spins and not simply between two spin $1/2$. The strong exchange couplings (20-30 K) between the eight spins of each ring rigidly lock them together in an entangled state (the $S=1/2$ state of each ring), which is preserved because of these strong couplings even in the presence of the sizeable magnetic field we are using in the experiment (see next point). This entanglement is connected to intra-ring correlations, which in turn reflect on large-Q modulations of the INS intensity (see for instance the intensity modulations as a function of Q_z in Fig. 2e in an ideal dimer). These two composite molecular spins are then entangled by the weaker inter-ring exchange (1.1 K) leading to the observed short-Q modulations. In addition, the entanglement between the rings in excited dimer states is fully preserved in applied fields.

Hence, 4D-INS allows us to "portray" not only the entanglement in the eigenstate of the supramolecular dimer, but also the real many-spin nature of the system. Information on the many spin nature of the dimer is completely lacking in the low-T susceptibility measurements we performed in Ref. [23] (now Ref. [28]), because the spatial information (the Q dependence) is not present in such technique. Indeed, at low T the susceptibility of the system is identical to that of a real dimer. Hence, this is an important strength point of the present technique.

We have rewritten most of the Introduction (lines 53-100) and the first part of the Results subsection "The $(Cr_7Ni)_2$ supramolecular dimer" (lines 103-147). In particular, in the Introduction the text has been completely revised in order to highlight the novelty and the most important aspects of our work. More details about molecular nanomagnets and the investigation of entanglement in these systems have been moved to the Supplementary Information, in order to shorten the general introduction and give the main message of the paper more immediately.

We thank the Reviewer for this remark, as we believe the paper is far better as a result, because we have now expressed more clearly important aspects of our work.

b) "Therefore one can use a magnetic field to place the system into a factorised 00 ground state and test the occurrence of entanglement in a more comfortable parameter space of (T,B) ."

The Referee is right. As stated in the previous point, the strong intra-ring exchange couplings rigidly lock the eight spins together in an entangled $S=1/2$ state. Since this condition is preserved also in sizeable fields, we can apply a field of 2.5 T to work in a regime that is comfortable for two reasons:

- 1) With an applied magnetic field the ground state is the factorized 00 state. In this way, we

can univocally attribute the observed inter-ring correlations to the entanglement in the excited state. Otherwise, if both states involved in the inelastic transitions were entangled states, the information would be mixed up.

- 2) By applying a field, we have improved significantly the experimental working conditions. Indeed, it is easier to resolve a very small splitting $\Delta = 0.04$ meV between two peaks in the center of the energy-transfer range, than to observe a peak centered at Δ , because it would be partially covered by the elastic signal.

We have now discussed more clearly in the paper our choice of working in an applied magnetic field, by adding a new detailed explanation in the first Results subsection (lines 148-173).

c) "why do the authors want to verify the occurrence of entanglement in weakly coupled antiferromagnetic rings forming a dimer by means of inelastic neutron scattering (INS) if entanglement has been already well characterised in this compound and published in ref. 23 ? Why do the authors try to verify that in a regime of magnetic fields (2.5T) and temperature (1.2K) where concurrence should be zero, as it was stated in ref. 23 ? And if so what have they measured in their INS ?"

We agree with the Referee that these points were not sufficiently discussed in the previous version of the paper. Here we are using $(\text{Cr}_7\text{Ni})_2$ as a benchmark to demonstrate the capability of this approach to study entanglement between generic molecular qubits. To test our method, we had to exploit a compound similar to one where we had already demonstrated the occurrence of entanglement with other techniques. However, this method can be applied also to dimers of more complex molecular qubits, where the full Hamiltonian is not known. Indeed, to demonstrate from the INS data the occurrence of entanglement (and to quantify it) we do not need to make any assumption on the form of the inter-qubit interaction. Moreover, as discussed above, 4D INS is also sensitive to the many spin nature of the molecular qubits.

At last, it is worth to underline that in this work we are detecting and quantifying the entanglement in the eigenstates of the system and not the entanglement in the thermodynamic equilibrium state (the canonical density matrix) as in old Ref [23] (now Ref. [28]). Here, we obtain very detailed information and we might extract the concurrence in the thermodynamic equilibrium state as a byproduct of this approach.

All these issues are now discussed in the new version of the Introduction (lines 76-101). We now also underline the generality of our approach in Discussion section (lines 390-400).

d) "In conclusion this manuscript is a very interesting piece of work that certainly deserves to be published in Nature Communications albeit with substantial refurbishment and improved writing. I strongly advise the authors to
- Make use of the Supplement Information
- State clearly why this work is important and hence preventing the reader to go back to original publications to figure it out. This is the main caveat of this manuscript."

We have strongly modified the Introduction and the subsequent part of the paper to make the focus and the novelty of the work clearer and we have added more than two pages in the Supplementary Information, where we have included more information about molecular nanomagnets and the investigation of entanglement in these systems. We believe these major revisions markedly improve the readability of the paper.

e) “ - In page 5 it has been made reference to the Ising model. Authors, could you please include such calculations in the Supplementary Information in order to strengthen their case ? “

We have added these calculations to the Supplementary Information as requested.

Reviewer #2:

We are grateful to the Reviewer for his comments on the paper, which helped us to better clarify important aspects of our work. Please find below a point-by-point response to each comment. A complete list of all the manuscript revisions is reported at the end of the document.

a) *“What I find a bit less clear is the discussion of the results suggesting that the presented results are a direct measurement of the entanglement between the molecular rings which is independent of the details of the models used to describe them. After careful reading the true extent of the claim becomes apparent, namely that it is only the entanglement between individual spins on the monomer rings that is being directly measured. Entanglement between monomers is proven only after invoking a model.”*

We agree with the Referee that are correlations between individual spins that are actually probed by INS. Nevertheless, dynamical correlations between spins belonging to different monomers would be zero if the states of the two monomers were factorized in the probed excited eigenstate (the sizeable field guarantees a factorized ground state), because the corresponding products of spin matrix elements would be zero. Hence, we can state that the observation of modulations in Q due to inter-ring correlations demonstrates that in the excited state the two rings are entangled.

For instance, in the case of Ising inter-qubit interaction the two monomers are not entangled and therefore the observed short Q modulations disappear (see the new Fig. S2). An effective Ising interaction could be obtained by coupling two monomers characterized by a strong easy-axis anisotropy.

In the revised version of the manuscript we have better clarified these points (see the changes in the Results subsection “Portraying entanglement with 4D-INS”, lines 195-208) and added the Figure S2 in the Supplementary Information.

b) *“Therefore the authors should discuss the following remarks and restate their claims to make this point clearer:*

1) The claim at the very end of introduction that the concurrence of the state of two spin is directly quantified is dependent on the validity of the models used in the analysis. Even though the used spin models are supported by many measurements and fit the data well it is not possible to claim that the measurement of the concurrence is direct. Similar issue appears in the analysis of Eq. 6, and the term $I_{\{AB\}}$ in it. Though the model with high concurrence shows the best fit to the observed data, the interpretation still depends on the facts that the monomer rings have total spin-1/2 states both in 0 and in p state. Therefore the language of the claim should explicitly state this.”

We agree with the Referee, the quantification of entanglement between molecular qubits requires some assumptions. In particular, in the case of $(Cr_7Ni)_2$ the main assumption is that each ring behaves at low temperature as a total spin 1/2. We now state this clearly in the introduction and

results sections of the revised version of the paper.

Anyway, the published measurements on the single-ring compound show that the ground state is a $S = 1/2$ doublet and that at the temperatures used here these are the only states of the ring that we need consider. In addition, molecular qubits can be always described as pseudospin $1/2$ (because they need to behave as two-level systems) and the method discussed in this paper can be extended to a generic doublet.

In the revised version we now explicitly state that we are assuming the $S = 1/2$ ground state of the rings, and we clarify that projection coefficients (Eq. 6 and subsequent) are here calculated by assuming the model interpreting measurements on the single-ring compound. However, these coefficients can be extracted from suitable measurements (previously this point was only discussed in note [38], which was note [37] in the previous version of the paper).

We have revised accordingly the Introduction (see, e.g. lines 70-75) and the Results subsection entitled "Quantification of entanglement between molecular qubits" from line 338. We have also made other changes in the Abstract, Introduction and Discussion sections. In particular, we removed "directly" in the points requested by the Referee.

It is also worth to underline that all the analysis in the second part of the paper is performed without any assumption on the form of the qubit-qubit interaction. We have now made this point more explicit (see Abstract, Introduction and Discussion sections).

c) *"2) In the scattering intensity plots, the structure that is claimed to appear due to entanglement of monomers is in fact due to the correlation of spins at relatively large distances. Therefore it can only be claimed directly that the oscillatory structure is due to entanglement between elementary spins, on Cr and Ni atoms, that lie on different monomers. It is not clear that the entanglement is between the frozen spins $1/2$ and it is not clear that the concurrence of the state can be found until the assumption of total spin $1/2$ on each of the rings is invoked. Therefore the claim of high concurrence is dependent on the model, and it should be stated so."*

Please see the previous two points. We have modified the paper to make the assumptions underlying the claim clear. In particular, we now explicitly state that the quantification of the entanglement is performed by assuming an $S=1/2$ ground state for the rings (lines 338-341).

d) *"The plots in Fig. 2 seem to have wrong labels. The plots do not correspond to the figure caption and the description in the main text."*

We apologize for this mistake. We have corrected the plots and figure captions so that they now match.

e) *"There is s mention of the states with artificially "turned off" correlations. The procedure for turning off the correlations should be explained explicitly."*

We have simply set to zero the inter-ring dynamical correlations in the cross-section formula and then calculated the INS response. We have now removed the phrase "turned off" from the text as it

was confusing and explained the procedure explicitly. In addition, we have now also considered the case of an Ising interaction as an interesting example of not-entangled rings.

We have now added the explanation in the main text and discussed the not-entangled Ising case in the Supplementary information.

Manuscript revisions

Abstract

Lines 16-20:

- we have changed “...to **directly** quantify it.” into “...to quantify it”: indeed the quantification of entanglement between molecular qubits **requires some assumptions** (i.e. in our work the main assumption is that each ring behaves at low temperature as a total spin $\frac{1}{2}$)
- we have changed “..., which allows **us**,...” into “which allows **one**” and “...,in eigenstates of **this** dimer” into “in eigenstates of **a** dimer” to underline the **universality of our approach**.

Introduction

Lines 29-44:

- we have changed “... and quantify entanglement **experimentally**...” into “quantify entanglement” and “...,**direct** experimental measurements...” into “experimental measurements”. We have changed these sentences to underline that with our approach the quantification of entanglement is indeed obtained from the comparison with experimental data, but **it requires some assumptions** (i.e. that each ring behaves at low temperature as a total spin $\frac{1}{2}$)
- we have removed the whole sentence “Among the most used techniques, there are...” , keeping all the References (13-19), in order to **shorten the general introduction and to give the main message** of the paper more immediately.

Lines 45-52:

- we have **added Reference 21**, linked to a State of the Art about molecular nanomagnets and entanglement in the **Supplementary Information**;
- we have **removed** the sentence “MNMs are magnetically-isolated spin clusters ...”: all the details about molecular nanomagnets **are now in the Supplementary Information**.
- we have **moved** the sentence “A large number of such complexes are now being synthesized,...” to the last part of the Introduction section (lines 88-94).

In lines 53-86 and 93-1010 the text has been completely revised in order to **highlight the novelty and the most important aspects of our work**. We introduce in more detail the potential of 4D-INS technique in detecting and quantify entanglement and we explain that this technique allows one to detect both intra- and inter-ring correlations. We illustrate why we have choose $(\text{Cr}_7\text{Ni})_2$ dimer as a benchmark to investigate entanglement in complex spin systems, a dimer made of two linked Cr_7Ni rings behaving as molecular qubit (i.e. the strong intra-ring exchange lead to an entangled $S = 1/2$ ground state for each monomer). We also explain the fundamental and experimental reasons for our in-field configurations to perform the 4D-INS experiment and the universality of our approach.

Results - The $(\text{Cr}_7\text{Ni})_2$ supramolecular dimer

Lines 103-147:

- we have **removed** the sentences:
 1. "In addition, excited states can be exploited as..."
 2. "...and can be magnetically linked in different ways in supramolecular structures."
 3. "They can also be grafted onto surfaces..."

More details about the Cr_7Ni AF ring are **now in the Supplementary Information**;

- we have **added** the sentence: "...and they have been proposed as prototype for implementing quantum gates.", to **underline the importance of Cr_7Ni AF rings** in the current research on **quantum information processing** with molecular nanomagnets.
- we have **added an important clarification** in lines 120-125: "This characterization shows, unequivocally, that the ground state is a $S = 1/2$ doublet...", which **was not present in the previous version** of the paper.
- We have **changed** the text from line 137 to line 143, keeping equation (2).

Lines 148-173:

- We have added a **more detailed discussion** about the **in-field configuration** of our 4D-INS experiment, which **was not present in the previous version** of the paper. By applying a magnetic field we can **induce a factorized ground state** for the dimer as a reference state to study INS transitions towards entangled states, also **improving the experimental conditions**.

Results - Portraying entanglement with 4-D INS

- We have **added** the sentence in lines 199-202: "Conversely, these inter-ring correlations would be zero if the states of the two monomers were factorized...", which allows us **to**

better explain the effect of inter-ring correlations reflect onto the scattering function, allowing one to “portray ” the entanglement between molecular qubits.

- We have correct the **labeling** of the panels of Fig. 2 in the text.
- In lines 262-264 we have **corrected** “...if correlations between the two rings are “**turned off**”...” into “...if correlations between the two rings are **forced to zero in Eq. 3...**”

Caption of Fig.2

- We have correct the **labeling** of the panels.

Results - Quantification of the entanglement between molecular qubits

From line 338:

- we have **added more details** about our **determination of I(Q)** functions in equation 6: we have **changed** the sentence “can be preliminarily determined from measurements on single-qubit compounds and are known for Cr₇Ni” into “are here **calculated by assuming the S = 1/2 doublet** deduced from the model interpreting measurements on the single-ring compound”. In this way we explicitly state that **we are assuming the S = 1/2 ground state of the rings.**
- We have also **added** the sentence: “It is important to note that these quantities can be also determined from measurements on single-qubit compounds”, to introduce note [38], to underline that projection coefficients are **here calculated from a model**, but that **they can be extracted from suitable measurements.**
- **Note [38]** (previously note [37]) has been simplified, since the most important information on the determination of I(Q) functions are **now in the main text.**

Discussion

- Lines 373-377: we have changed “...to **directly** quantify it.” into “...to quantify it”;
- Lines 390-396: we have better explained the **universality of our approach**, by **adding the sentences**: “It is important to underline that the present method works independently of the specific form of the inter-qubit interaction. Indeed, neither the demonstration of the entanglement through the observed short-Q modulations of the intensity, nor its quantification using equation (6), exploit equation (2).”.

Supplementary Information

More than **two pages** have been added to the Supplementary Information:

- We have **added more information** about molecular nanomagnets and the investigation of entanglement in these systems
- We have **added** the study of a $(\text{Cr}_7\text{Ni})_2$ dimer **with inter-ring Ising interaction** to demonstrate that short-Q modulations disappear when the eigenstates of the dimer are not entangled.

REVIEWERS' COMMENTS:

Reviewer #1 (Remarks to the Author):

I am satisfied with the modifications that the authors have carried out in their manuscript and this reviewer recommend this manuscript for publication as it is.

Reviewer #2 (Remarks to the Author):

In the revised manuscript, and in response to earlier comments, the authors have clearly stated the assumptions behind their reasoning. In addition, the revised manuscript is more to the point and clearly describes the results. I would like to recommend it for publication.

Reviewer #1

"I am satisfied with the modifications that the authors have carried out in their manuscript and this reviewer recommend this manuscript for publication as it is."

We thank the Reviewer for his favourable comments. We are also grateful for his careful review and suggestions, which contributed to improve the quality of our manuscript.

Reviewer #2

"In the revised manuscript, and in response to earlier comments, the authors have clearly stated the assumptions behind their reasoning. In addition, the revised manuscript is more to the point and clearly describes the results. I would like to recommend it for publication."

We thank the Reviewer for his favourable comments. We are also grateful for his careful review and suggestions, which contributed to improve the quality of our manuscript.